# Silencing of topical proline hydroxylase domain 2 promotes the healing of rat diabetic wounds by phosphorylating AMPK

**Defu Xie[1,2], Mengchang Liu[1,2], Yingxi Lin[1,2], Xingke Liu[1,2], Hong Yan[1,2]***

**1** Southwest Medical University, No. 1 Section 1, Xianglin Road, Luzhou City, Sichuan Province, 646000, China, **2** National Key Clinical Construction Specialty, Wound Repair & Regeneration Laboratory, Department of Plastic & Burn Surgery, Affiliated Hospital of Southwest Medical University, No. 25 Taiping Street, Jiangyang District, Luzhou,Sichuan Province, 646000, China

* xnshaoshang123@163.com

## Abstract

### Background

For diabetic ulcers, the impaired response to hypoxia is a key feature associated with delayed healing. In the early phase of hypoxia, hypoxic signaling activates the AMPK system through direct phosphorylation of the PHD2 pathway, producing a significant endogenous hypoxic protective effect.

### Methods

Twenty Sprague-Dawley (SD) rats were randomly divided into two groups: treatment (sh-PHD2) and control (sh-Control). Using lentiviral encapsulation of PHD2-shRNA and transfection, the silencing efficiency of PHD2 expression was verified in rat dermal fibroblasts (RDF) and in rat aortic endothelial cells (RAECs). Changes in the ability of RDF and RAECs to proliferate, migrate, and in the rate of ATP production were observed and then tested after inhibition of AMPK phosphorylation using dorsomorphin. The lentiviral preparation was injected directly into the wounds of rats and wound healing was recorded periodically to calculate the healing rate. Wounded tissues were excised after 14 days and the efficiency of PHD2 silencing, as well as the expression of growth factors, was examined using molecular biology methods. Histological examination was performed to assess CD31 expression and therefore determine effects on angiogenesis.

### Results

Lentiviral-encapsulated PHD2-sh-RNA effectively suppressed PHD2 expression and improved the proliferation, migration, and ATP production rate of RDF and RAEC, which were restored to their previous levels after inhibition of AMPK. The rate of wound healing, vascular growth, and expression of growth factors were significantly improved in diabetic-model rats after local silencing of PHD2 expression.

### Conclusion

Silencing of PHD2 promoted wound healing in diabetic-model SD rats by activating AMPK phosphorylation.

**Data Availability Statement:** All relevant data are within the paper and its Supporting Information files.

**Funding:** The authors who received the award is Mr Yan. The full name of the funder is (Luzhou-Medical University) Cooperation Project,the grant number is No.2019LZXNYDZ08.The URL of funder website is http://luzhou.tccxfw.com. The funders had no role in study design, data collection and analysis, decision to publish, or preparation of the manuscript.

**Competing interests:** The authors have declared that no competing interests exist.

## Introduction

The latest data from the International Diabetes Federation (IDF) show that in 2017, there were approximately 425 million patients diagnosed with diabetes in the world, of which 114 million were located in China, ranking first in the world [1]. Microvascular and macrovascular lesions are one of the main complications in patients with diabetes and are considered an important risk factor for death [2, 3]. It is also an intrinsic cause of impaired wound healing [4, 5]. This is because the high glucose environment and the local inflammatory environment of the wound can lead to a decrease in the number and function of endothelial progenitor cells, thus downregulating the expression of vascular endothelial growth factor (VEGF) and its receptors and inhibiting the formation of new blood vessels [6, 7]. In addition, vascular rupture and the inflammatory response increases the oxygen consumption of tissue cells, further worsening hypoxia in the injured area. Therefore, for patients with diabetes, one of the key factors for wound healing is whether they can timely and effectively respond to the hypoxic state.

Prolyl hydroxylase domain protein 2 (PHD2) is widely recognized as a fine oxygen sensor, a central regulator of cellular oxygen homeostasis, and a major inducer of the adaptive response to hypoxia. In the presence of oxygen, PHD2 hydroxylates an oxygen-sensitive in vivo-specific proline residue HIF-1α, which is then marked for degradation by the proteasome [8]. However, under hypoxic conditions, the catalytic activity of PHD2 hydroxylation decreases, and thus maintains the stability of HIF-1α and activates the transcription of its target genes in the nucleus. However, current studies have shown that not only does PHD2 play the role of an "HIF regulator" in the oxygen detection signal cascade, it also includes other protective anti-hypoxia responses that do not depend on HIF [9, 10]. More critically, activation of the PHD2 signaling pathway modulates various cellular responses to hypoxia and preconditioning stimuli [9]. Under hypoxic conditions, cell survival depends on the ability to efficiently maintain intracellular ATP levels in the presence of restricted mitochondrial oxidative metabolism and the lack of bioenergetic substrates. Therefore, the state of cellular energy is a decisive factor in cell survival and maintaining cellular metabolic homeostasis under hypoxia is particularly important for promoting cellular biological activity. AMP-activated protein kinase (AMPK) is a master regulator of metabolic homeostasis and a cellular energy sensor. It is a heterotrimeric serine/threoninase consisting of 1 catalytic subunit (α1 or rα2) and 2 regulatory subunits (β1 or rβ2 and dγ1, γ2, or γ3) [11, 12]. AMPK is activated under conditions of elevated AMP/ATP, induced by glucose deprivation, muscle contraction, and hypoxia [13], which lays a critical role in maintaining cellular energy homeostasis and adaptive responses [14]. After activation, it can directly phosphorylate downstream proteins or indirectly influence the expression of genes that reduce additional consumption of ATP and increase the tolerance of cells to hypoxia [15]. Combining the role of PHD2 in regulating cellular responses to hypoxic stimulation in the hypoxic state, we speculate that local silencing of PHD2 can promote wound healing in diabetic rats by activating the AMPK pathway.

## Methods

### Animals, cell lines, and lentiviral vector

Male SD rats (180 g) were purchased from the Southwest Medical University Experimental Animal Centre (Sichuan,China). The rats were housed in a standard environment at room temperature (24±2°C, relative humidity (50±5% =, and a light/dark cycle of 12-h/12-h. All animal operations were performed according to protocols approved by the Animal Care and Use Committee of Southwest Medical University. Methods of sacrifice: The animals were euthanised by intraperitoneal injection of 150 mg/kg of sodium pentobarbital. Efforts to alleviate

suffering: Through improving animal facilities, breeding management and experimental conditions, selecting experimental animals, technical routes and experimental means, optimising experimental operation techniques, minimising damage to the animal organism during the experimental process, and reducing the pain and stress suffered by the animals. Rat aortic endothelial cells (RAECs) were purchased from Meisen Chinese Tissue Culture Collections (Meisen CTCC, China). Rat dermal fibroblasts (RDF) were purchased from Procell Life Science & Technology Co., Ltd. (Wuhan, China). Cells were cultured in Dulbecco's Modified Eagle Medium(DMEM) (Gibco, USA) containing 10% fetal bovine serum, 100 IU/mL penicillin and 100 μg/mL streptomycin in an incubator at 37˚C with 5% $CO_2$. The medium was changed every 2 to 3 days. The lentiviral vector expressing rat PHD2-shRNA (pLV.U6.shRNA2(rPHD2).CMVcopGFP-2A-Puro.WPRE) was constructed by PackGene Biotech Co., Ltd. (Guangzhou, China). The lentiviral vector expressing only GFP (pLV.U6.NCshRNA-1. CMV.copGFP-2A-Puro.WPRE) was used as a control. The viruses were produced using Gene-Copoeia protocols (https://www.genecopoeia.com). According to the Gene-Copoeia protocol, lentiviral particles were generated using lipofectamine-mediated lentiviral-expressing plasmids, the packaging plasmid psPAX2 and the envelope plasmid pMD2.G in CAF, and lentiviral particles were collected to transduce target cells.

### Lentiviral vector transfection

Cells (RDF, RACE) were seeded in 6-well plates ($5 \times 10^4$ cells/well) and cultured overnight. When the cells reached 50–60% confluency, the culture medium was removed. The cells were then transfected with virus-containing diluted medium with 8 μg/mL polybrene for 4 h. Then, the medium was replaced with serum-enriched medium, and the cells were cultured for an additional 96 h. Transfected cells were pooled and treated with puromycin for 1 week. Subsequently, transfected cells were collected and processed for follow-up experiments. These cells were divided into three groups: sh-PHD2 (sh-PHD2 infection group), sh-Control (control group expressing GFP alone), and sh-PHD2+8 μM dorsomorphin (after sh-PHD2 infection, cells were treated with 8 μM dorsomorphin were treated for 24 hours). Dorsomorphin was purchased from AbMole (China) (Cat. No. M2238), a selective reversible inhibitor of AMPK.

### Scratch test

One day in advance, infected cells were inoculated in a six-well plate according to $2 \times 10^5$/well, and the 20 μL pipette tip was used to quickly draw a scratch along the middle of the plate. After washing with PBS, 2 mL of serum-free medium was added and the plates were imaged after 0, 6, 12, 24 h of cell migration with an inverted microscope to calculate the migration area.

### Detection of cell proliferation and cell viability using the CCK8 method

The proliferation of RDFs and RAECs was detected using the Shanghai Ethan Cell Counting Kit 8 (CCK8). Each group of cells was seeded in 96-well plates at 2000 cells/well. Each well contained 100 μL of complete medium and 10 μL of CCK8 reagent was added after 24, 48, 72 and 96 h, and the absorbance at 450 nm was measured after 4 h of incubation.

### Real-time ATP rate assay

The day before the experiment, cells were seeded in Seahorse XFe 24-well microplates at 15,000 cells/well, and the probe plates were hydrated overnight in a 37˚C $CO_2$-free incubator. On the day of the experiment, the detection solution was prepared in advance by adding 10

mmol/L XF (Extrancellular Flux) glucose, 1 mmol/L XF pyruvate, and 2 mmol/L XF glutamine to 100Ml Seahorse XF DMEM pH 7.4) and preheated to 37˚C. The microplate was then prepared by first washing the microplate with detection solution and then adding 500 μL of freshly preheated detection solution. Next, a 15 μmol/L oligomycin and 5 μmol/L the rotenone/antimycin A mixture were prepared and added to the appropriate wells. Absorbance was then read using the Seahorse XFe24 Analyzer (Agilent Technologies, Sana Clara, USA) according to the manufacturer's instructions.

## Preparation of the SD rat diabetic wound model

After 3 days of adaptive feeding, rats were randomly divided into two groups: the treatment group (sh-PHD2) (in the diabetes model rats, the wound was injected with the PHD2-shRNA lentivirus) (n = 10) and the control group (sh-Control) (in the diabetes model rats, the wound was injected with the sh-Control lentivirus) (n = 10). The diabetic rat model was established by a one-time injection of streptozotocin (50 mg/kg) into the tail vein (AbMole, China). The fasting blood glucose level of each rat was detected 72 h after streptozotocin injection: fasting blood glucose values were ≥16.70 mmol/L, and were indicative of the successful preparation of the diabetes model. After ensuring the diabetes model had been achieved, all rats (including the control group) were used to establish wound models. A circular area with a diameter of 2 cm was marked in the middle of the back, and the skin surface was prepared and disinfected. Using a scalpel and ophthalmic scissors, a full thickness skin defect was created on the marked area, and a sterile dressing was applied to the wound and surrounding areas after injecting the corresponding drugs.

## Wound healing rate

On days 0, 3, 7, 14, and 21, digital cameras were used to collect photos of back wounds under fixed focal length and pixel conditions, respectively. The wound area was calculated using special ImageJ software, and the degree of wound healing was calculated according to the change in the wound area using the following formula: wound healing rate = (wound area—wound area at each time point) / wound area × 100%.

## Western blotting analysis

To determine the level of expression of growth factors (VEGF, fibroblast growth factor 2 [FGF-2]) in wound tissue and the inhibition efficiency of PHD2 in the wound, samples of wound tissue and surrounding areas were collected on day 14. Protein expression was determined using mouse anti-rat PHD2 antibody (Abcam, USA) and mouse anti-rat VEGF antibody (Abcam, USA) and mouse anti-rat FGF-2 antibody (Abcam, USA).

## Reverse transcription polymerase chain reaction (RT-PCR)

Rat *PHD2*, *GAPDH*, *VEGF*, and *FGF-2* gene sequences were recovered from PubMed/Negyl GenBank, with GAPDH as the reference gene. Primers were designed using Primer Express 5.0 software as follows: PHD2 sequence: forward: 5'-TTGATAGACTGCTGTTTTTCTGG-3'; reverse: 5'-CCTCACACCTTTTTCACCTGTTA-3'. GAPDH sequence: forward: 5'-CCTGGAGA AACCTGCCAAGT-3'; reverse: 5'-TAGCCCAGGATGCCCTTTAG-3' [10, 16]. VEGF sequence: forward: 5'-GTCACCGTCGACAGAACAGT-3'; reverse: 5'-GACCCAAAGTGCTCCTCGAA-3'. FGF-2 sequence: forward: 5'-TCCATCAAGGGAGTGTGTGC-3'; reverse: 5'-GGACTCCAGGC GTTCAAAGA-3'. Total RNA extraction was performed after RAECs were successfully infected

and treated with puromycin for 1 week. mRNA quantification was performed by real-time quantitative RT-PCR.

## CD31 immunohistochemistry

On day 14 after injury, the wound surface and surrounding 2-mm thick uninjured skin were excised to prepare paraffin sections. Vascular endothelial cell density was evaluated by CD31 immunohistochemical staining. Rat liver sections were stained as the positive control group (PCG). Paraffin-embedded sections were prepared using normal rat wounds, and PBS was used instead of the primary antibody as the negative control group (NCG).

## Statistical analysis

Statistical analysis was performed using SPSS for Windows ver. 13.0 (SPSS, inc., Chicago, IL, USA).A two-sided P<0.05 was considered statistically significant and is represented in figures as *P<0.05, **P<0.01, ***P<0.001.

## Results

### PHD2 expression decreased in RAECs after infection with the PHD2-shRNA lentivirus

Fluorescence detection of PHD2 expression performed one week after virus infection to evaluated virus infection. After RAECs were successfully infected and treated with puromycin for 1-week, quantitative real-time fluorescence RT-PCR was performed to detect the mRNA expression of PHD2. After treatment, the expression of PHD2 mRNA decreased to 16.6%, indicating the effectiveness of virus-induced silencing (P<0.001) (Fig 1A).

### Western blotting analysis

The western blotting analysis showed that the relative density of PHD2 in the sh-PHD2 group was significantly lower than in the sh-Control group, and there was no significant change in PHD2 expression after the addition of dorsomorphin (P<0.05); differences in the remaining groups were not statistically significant. The difference in relative AMPK density between the groups was not statistically significant. Since AMPK is phosphorylated prior to exerting its functional activity, we further explored the expression of phosphorylated-AMPK (p-AMPK) and found that its relative density was significantly higher in the sh-PHD2 group (P<0.05), while the difference was not statistically significant in the remaining groups (Fig 1B). Using ImageJ quantitative analysis, the expression of PHD2 was reduced by 62.7% ± 2.3% in the experimental group, while AMPK phosphorylation levels increased by 47.95% ± 2.635% (Fig 1C).

### PHD2 enhanced RDF and RAEC cell proliferation and viability, and this effect disappeared when AMPK phosphorylation was inhibited

After four consecutive days of absorbance measurements, we compared the value-added rate of each group of cells for the following three days using the absorbance of the first day as a benchmark. Cell proliferation in the sh-PHD2 group was significantly higher than that in the sh-Control group, indicating that PHD2 silencing could enhance the proliferative capacity of both cell lines. The proliferative capacity decreased significantly after the addition of dorsomorphin (Fig 2A), which indicates that inhibition of AMPK phosphorylation can effectively inhibit the enhanced proliferative capacity brought about by silencing PHD2. Similarly, the

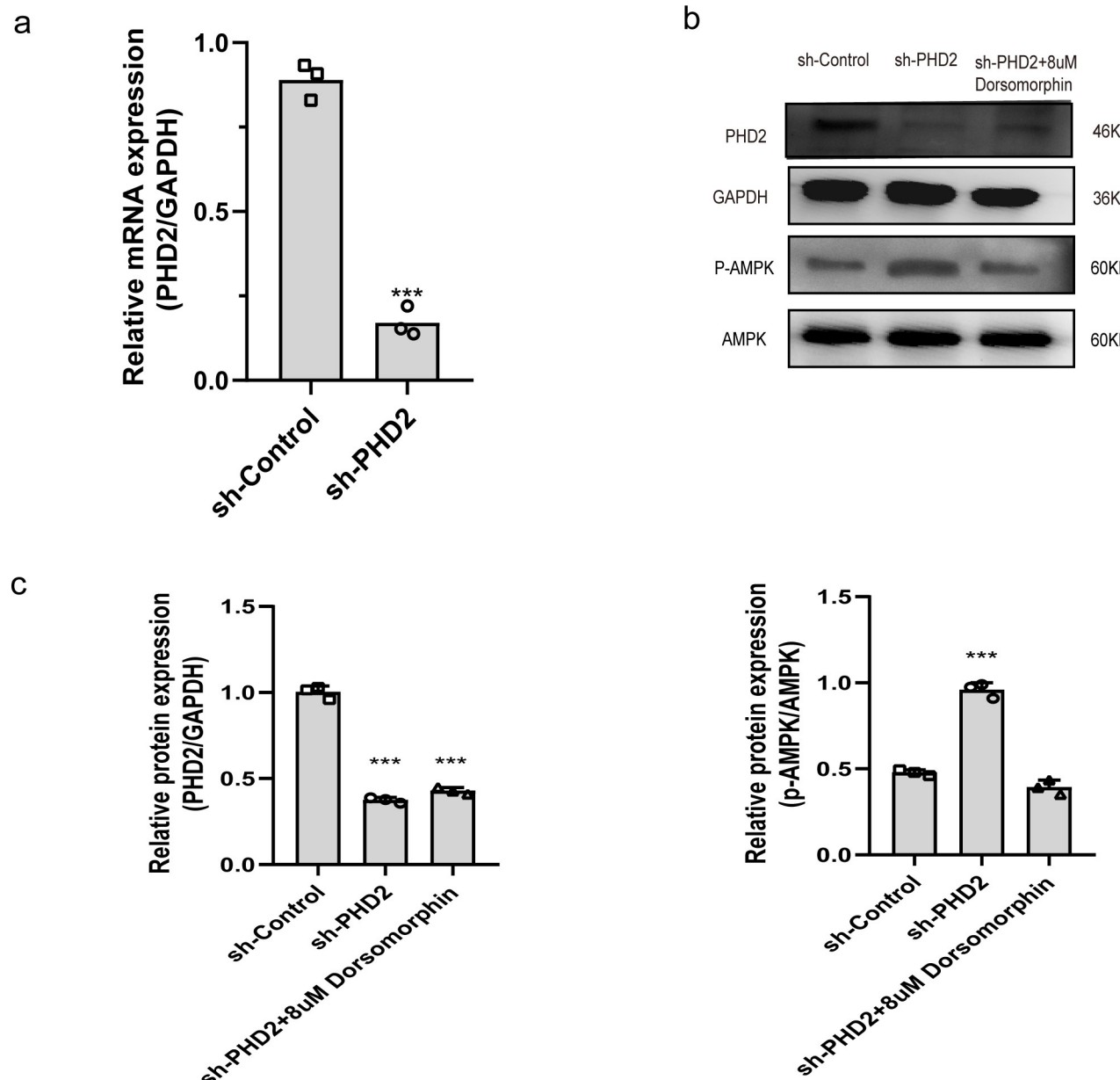

**Fig 1.** a:PHD2 mRNA expression was significantly reduced in RAECs after successful PHD2-shRNA infection compared with controls (p<0.001). b,c: Western-blot analysis of the expression levels of PHD2 protein and the phosphorylation level of AMPK protein in different groups.

absorbance of the cells in each group measured at 72 h, using the sh-Control group as a benchmark for cell viability, were consistent with the above trend (Fig 2B).

## PHD2 improved the migration capacity of RDF and RAEC, and this effect was abolished by inhibition of AMPK phosphorylation

Using scratch assays, the migration ability of RDFs in the PHD2-shRNA group was significantly enhanced after effective silencing of PHD2. Compared to the control group, the width of the scratch was significantly reduced at 12 and 24 h and the healing rate was significantly faster (P<0.05) (Fig 3A). We then quantified the migration ability of the RAECs using a

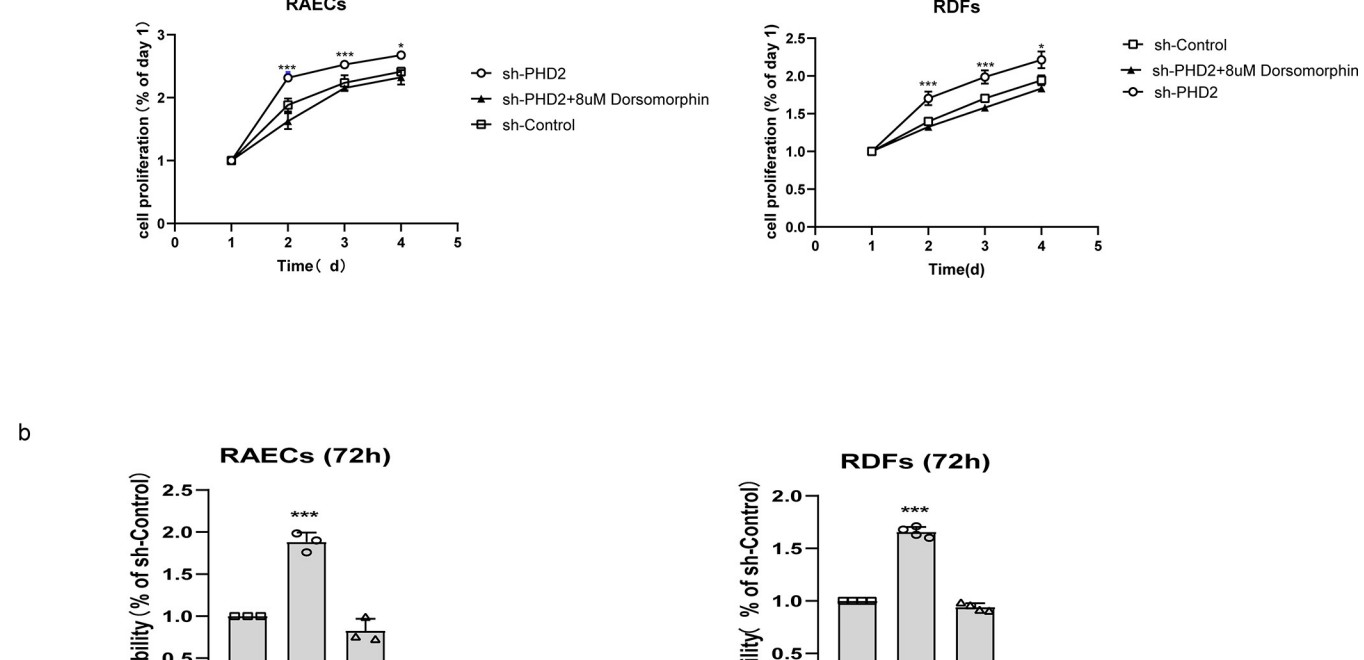

**Fig 2.** a: Changes in the proliferative capacity of cells in different groups at days 1, 2, 3, and 4. b: Changes in the viability of cells in different groups at 72h.

Transwell assay. The migration ability of endothelial cells in the PHD2-shRNA group was significantly higher than that of the control group and the difference was statistically significant (P<0.05) (Fig 3B). After inhibiting AMPK phosphorylation, the above-mentioned effects were significantly reduced.

## PHD2 enhanced the overall ATP production rate of RAEC and RDF cells, and the effect was lost when AMPK phosphorylation was inhibited

We used the Seahorse XFe24 extracellular flux analyzer to quantify real-time rates of ATP synthesis of RAECs and RDFs. The rate of glycoATP production represents the rate of ATP synthesis associated with the conversion of glucose to lactate via the glycolytic pathway, and the rate of mitoATP production represents the rate of ATP synthesis associated with mitochondrial oxidative phosphorylation, and these are summed to obtain the total ATP production rate. We found that the total rate of ATP production of both RAEC (Fig 4A) and RDF (Fig 4B) was significantly higher than that of the control group after effective silencing of PHD2, implying that the total metabolic capacity of both cells was enhanced. We then calculated the ratio of mito-ATP production to glyco-ATP production for both cell lines. The ratio in the Sh-PHD2 group was significantly lower than that in the Sh-Control group (Fig 4C and 4D), which indicated that the Sh PHD2 group achieved a higher glyco-ATP production rate, and also implied that its aerobic glycolytic capacity was improved.

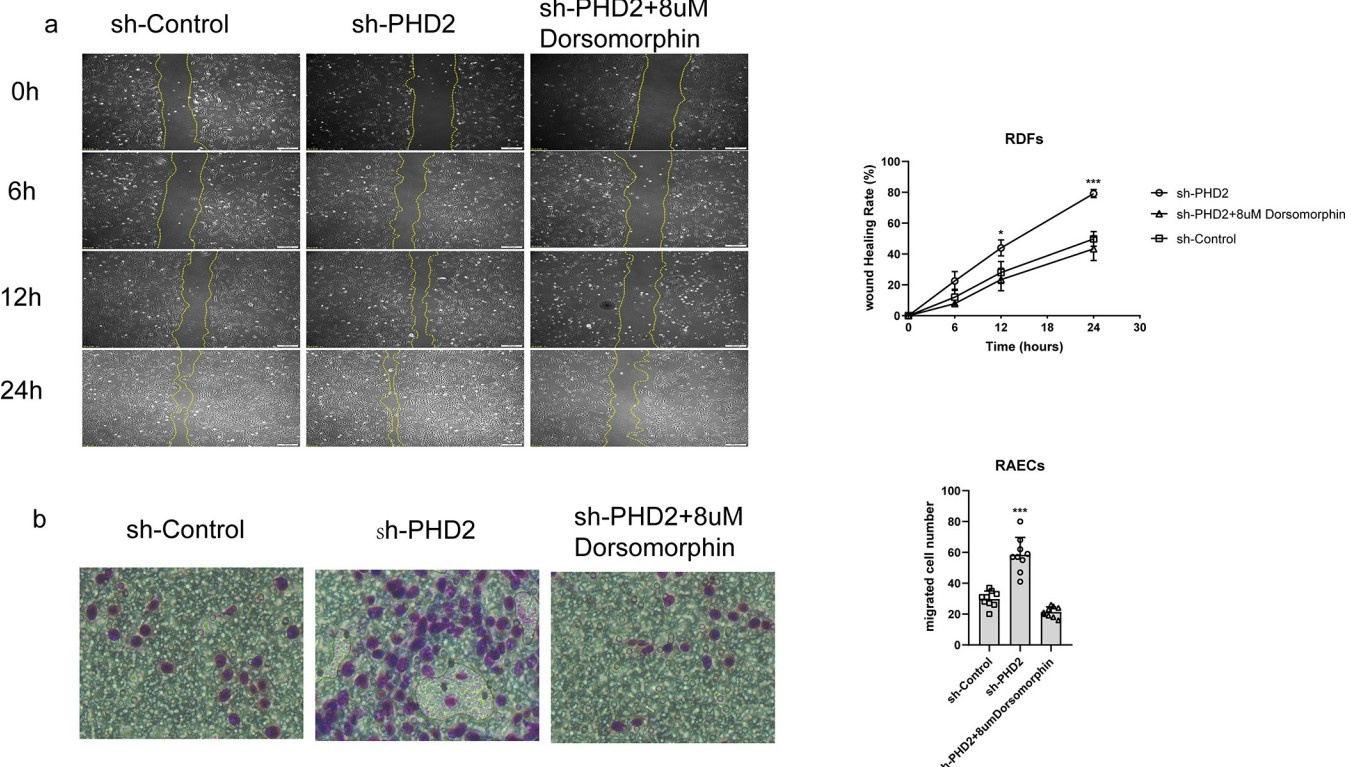

**Fig 3.** a: Scratch test was used to detect the migration ability of RDFs in different groups.b: Transwell experiment was used to detect the migration ability of RAECs.

### Topical PHD2 silencing promoted wound repair in diabetic rats

Gross observation of wounds on days 0, 3, 7, 10, 14, and 21 revealed a significant increase in wound healing rate in the treatment group (Fig 5A). From day 3 onwards, there was a significant increase in wound closure at each time point relative to the control group (Fig 5B), with wounds in the treated group starting to heal on day 7 (74%), while only 47% of wounds in the control group healed at the same time point, and this relationship continued until wound closure was achieved in the experimental group.

### Silencing of PHD2 expression promoted angiogenesis in wounds

Wound angiogenesis was characterized and quantified by the detection of CD31 by immunohistochemical staining. There was an increase in endothelial cell density in rat wounds following silencing of PHD2 on day 14 (Fig 6A). Quantitative analysis of the staining results of the rat wound sections using Image J showed that the staining density of CD31 increased by $26.13\% \pm 2.38\%$ in the experimental group compared to the control group (Fig 6B). These results suggested that silencing of PHD2 exerted a promotional effect on angiogenesis in the diabetic rat wound model.

### Western blot analysis of tissues

Western blotting analysis revealed that the relative density of PHD2 was significantly lower in both treatment groups on day 14 ($P<0.05$), while the relative densities of both VEGF and FGF-2 increased significantly ($P<0.05$) (Fig 7).

a

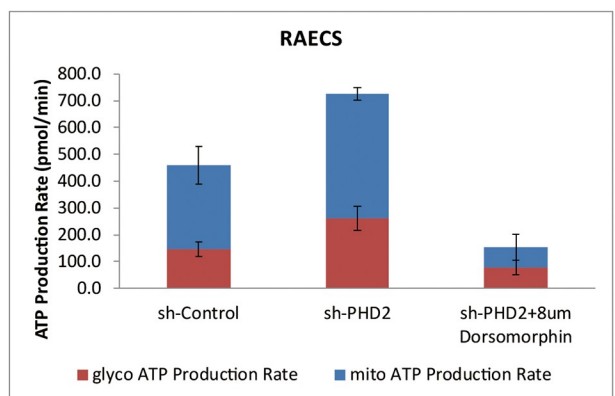

c

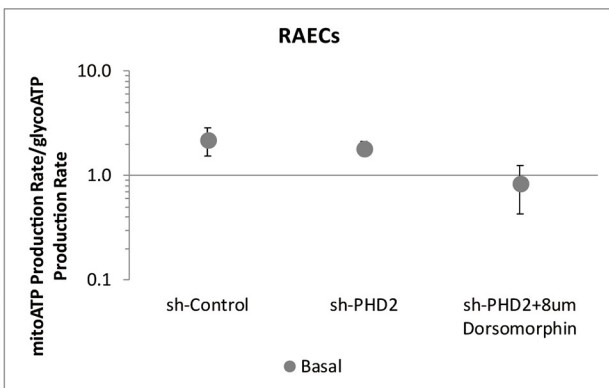

b

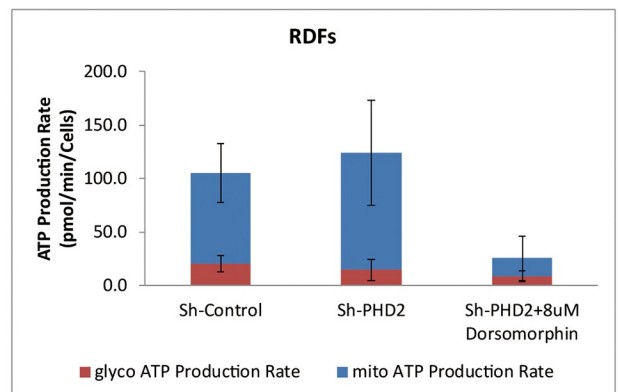

d

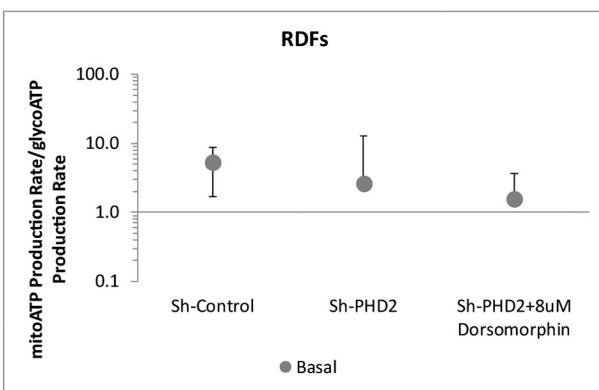

**Fig 4.** Total ATP Production Rate of RAECs (a) and RDFs (b) and the ratio of mito-ATP Production Rate to glyco-ATP Production Rate of RAECs (c)and RDFs(d).

a

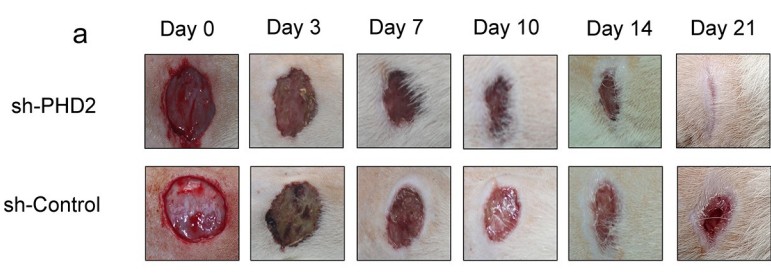

b

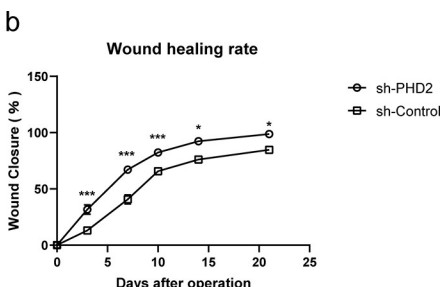

**Fig 5.** a: Wound healing of treatment group (sh-PHD2) and control group (sh-Control) on days 0, 3, 7, 10, 14, and 21.b: Wound healing rates of two groups of rats at different time points.

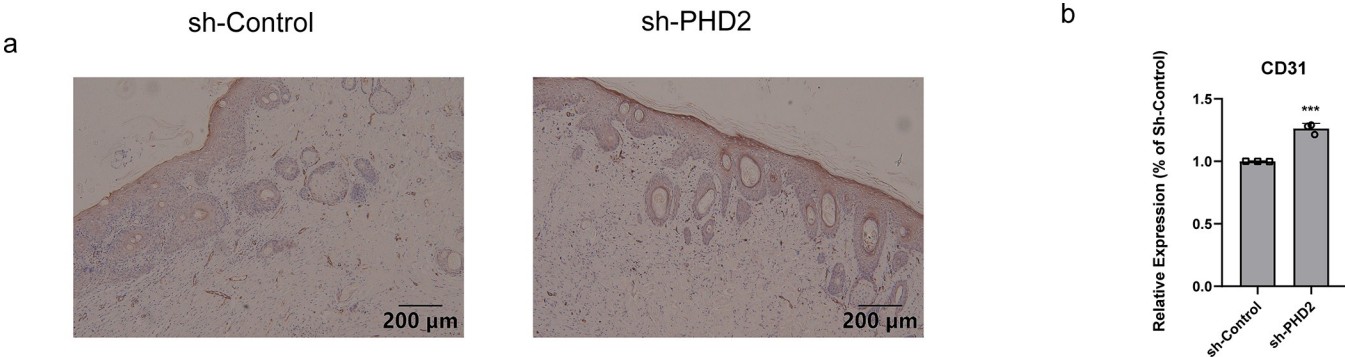

**Fig 6.** a,b: Results of traumatic tissues of two groups of animals on day 14 after staining for CD31 by immunohistochemistry and after quantification of their CD31 expression levels.

## Discussion

For patients with diabetes, the neovascularization ability of a wound is significantly altered, thus diabetic wounds are undoubtedly hypoxic [17–21]. The proteolytic hydroxylase domain (PHD1-3) is an oxygen-sensitive enzyme that acts as an oxygen sensor in the cell, providing a direct link between oxygen availability and cellular adaptation to hypoxia. Among the three distinct members of the PHD family, PHD2 has been shown to be the most abundant and critical hydroxylase in most cell lines studied to date [22]. Although its most recognized role is that of a 'HIF regulator', we previously found that PHD2 can activate calmodulin-dependent protein kinase (CaMKK) β upstream of AMP-activated protein kinase (AMPK) and thereby protect against hypoxia-induced effects in cardiomyocytes; phosphorylated AMPK is a key factor in cell survival as it regulates energy metabolism [15, 23]. This hypothesis has also been confirmed by other researchers studying systemic iron homeostasis [24]. Thus, we speculated that in diabetic wounds, PHD2 may promote wound repair by regulating AMPK. Therefore, we first performed local silencing of PHD2 expression in wound tissue of diabetic-model rats using lentivirus coated PHD2-shRNA, and the efficiency of wound repair increased significantly, as expected. We then quantified angiogenesis by evaluating the presence of CD31-labeled vascular endothelial cells [25]. Our results also indicated that silencing PHD2 was beneficial for stimulation of angiogenesis within the wound. To demonstrate the effectiveness

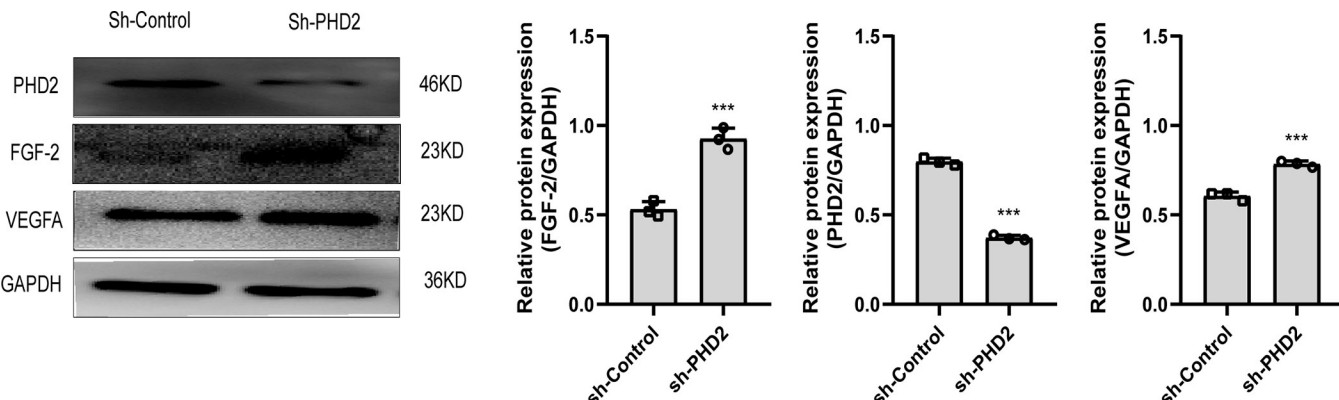

**Fig 7. The expression levels of PHD2, VEGFA, and FGF-2 in the wounds of the two groups of rats were analyzed by Western-blot.**

of local silencing in the wound region, we also performed a quantitative analysis of wound tissue proteins and found that the density of PHD2 expression in the treatment group was significantly lower than in the control group. The growth factors VEGF and FGF-2 have a synergistic effect on angiogenesis and are significantly more effective than cytokines alone [26–29]. None of the clinical trials to date testing VEGF or FGF-2 treatments alone have demonstrated significant clinical efficacy [27, 30, 31]. We also quantified VEGF and FGF-2 levels, both of which showed a marked increase expression in tissues in harboring cells without PHD2 expression.

Having determined that silencing of PHD2 in wound tissue is beneficial for wound repair, we next needed to explore whether activation of AMPK was also involved in this response. Due to the rapid activity of AMPK inhibitors on the wound surface and the instability of phosphorylated-AMPK itself, we selected two cell lines that were closely related to wound repair to discuss this mechanism. Impaired wound healing in patients with diabetes is a well-known phenomenon, and fibroblast dysfunction may be involved in this process [32–34], while endothelial cells are an integral part of angiogenesis. We first demonstrated the effectiveness of inhibition of PHD2 expression, as demonstrated by the RT-PCR and western blotting results following lentiviral silencing PHD2. Next, we determined that silencing of PHD2, was associated with a significant increase in the phosphorylation of AMPK, and this effect could be completely reversed by exposure to dorsomorphin (an AMPK inhibitor) without altering PHD2 expression. Based on this evidence, we explored changes in the proliferation and migration abilities of these two types of cells and found that cells in which the expression of PHD2 was silenced, there was an enhancement of proliferation and migration, with the effects disappearing completely following exposure to dorsomorphin.

It is well known that AMPK is the main regulator of metabolic homeostasis and as a cellular energy sensor, it is the guardian of the energy state of the cell [24, 35]. AMPK is a key regulator of energy homeostasis, and coordinates adaptive responses in metabolic states depleting ATP such as hypoxia, ischemia/reperfusion, and exercise [36]. Therefore, to explore the processes through which energy changes are necessary for living organisms, we further demonstrated that silencing of PHD2 could enhance the total ATP production rate of these two cells, and the effect can still be inhibited by AMPK inhibitors. Furthermore, the ratio of mito-ATP production to glyco-ATP production was calculated to confirm that the aerobic glycolytic capacity, which showed that in both cell lines ATP production was also significantly enhanced after silencing of PHD2.

Although we found that a decrease in PHD2 expression promotes an increase in the phosphorylation level of AMPK, it is undeniable that it also brings about alterations in the expression of other proteins, the most classical and widely recognised of which is the regulation of hif-1α. When oxygen is sufficient, in the co-presence of iron and 2-oxoglu- tarate (2-OG), PHD2 can hydroxylate specific proline residues (Pro402, Pro564) [37] and asparagine residues of hif-1α, which leads to their binding to von HippelLindau protein (pVHL)-E3 ubiquitin ligase complex and finally to proteasomal degradation mediated by E2 ubiquitin ligand [38]. Thus, when PHD2 expression is reduced, hif1-α is unquestionably increased in vivo and the physiological responses it dominates are enhanced. Current studies have shown that the genes downstream of hif-1α mainly include *vascular endothelial growth factor (VEGF), erythropoietin (EPO)*, GLUT1 glucose trans-porter, glycolytic enzymes [39]. The increased expression of these genes also had a positive effect on diabetic wound repair, which suggests to us that perhaps there is a synergistic effect between the increased phosphorylation of AMPK and the increased expression of hif-1α, and we will also further study and investigate the relationship between the two pathways afterwards.

## Conclusions

Local silencing of PHD2 in wound tissue can activate a series of adaptive responses of hypoxia-diabetic wounds via phosphorylation of AMPK, thus promoting angiogenesis and accelerating wound repair.

## Supporting information

**S1 File.**
(ZIP)

**S2 File.**
(ZIP)

**S3 File.**
(ZIP)

**S4 File.**
(ZIP)

**S5 File.**
(ZIP)

**S6 File.**
(ZIP)

**S7 File.**
(ZIP)

**S8 File.**
(ZIP)

**S1 Raw images.**
(PDF)

## Acknowledgments

The pilot research was conducted in National Key Clinical Construction Specialty, Wound Repair and Regeneration Laboratory.

## Author Contributions

**Conceptualization:** Defu Xie, Hong Yan.

**Data curation:** Defu Xie.

**Formal analysis:** Defu Xie, Mengchang Liu.

**Funding acquisition:** Hong Yan.

**Methodology:** Defu Xie, Mengchang Liu, Yingxi Lin, Xingke Liu.

**Project administration:** Defu Xie.

**Writing – original draft:** Defu Xie.

**Writing – review & editing:** Defu Xie.

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
