## [Decision Letter · Decision Letter 0]

31 Aug 2023

PONE-D-23-10650Silencing of topical proline hydroxylase domain 2 promotes the healing of rat diabetic wounds by phosphorylating AMPKPLOS ONE

Dear Dr. Yan,

Thank you for submitting your manuscript to PLOS ONE. After careful consideration, we feel that it has merit but does not fully meet PLOS ONE’s publication criteria as it currently stands. Therefore, we invite you to submit a revised version of the manuscript that addresses the points raised during the review process.

We look forward to receiving your revised manuscript.

Kind regards,

Kanhaiya Singh, Ph.D

Academic Editor

PLOS ONE

Journal Requirements:

2. To comply with PLOS ONE submissions requirements, in your Methods section, please provide additional information regarding the experiments involving animals and ensure you have included details on (1) methods of sacrifice, (2) efforts to alleviate suffering.

"Tthe authors who received the award is Mr Yan . The  full name of the funder is (Luzhou-Medical University) Cooperation Project,the grant number is No. 2019LZXNYDZ08.The URL of funder website is http://luzhou.tccxfw.com .The sponsor was involved in the design of this study."

6. Please upload a copy of Supporting Information Figure/Table/etc. Fig.1-7 which you refer to in your text on page 22.

Additional Editor Comments:

Although the reviewers have found this study of interest, they have suggested to have additional validation of the data presented.

Reviewers' comments:

Reviewer's Responses to Questions

**Comments to the Author**

1. Is the manuscript technically sound, and do the data support the conclusions?

Reviewer #1: Yes

Reviewer #2: Yes

2. Has the statistical analysis been performed appropriately and rigorously? 

Reviewer #1: Yes

Reviewer #2: Yes

3. Have the authors made all data underlying the findings in their manuscript fully available?

Reviewer #1: Yes

Reviewer #2: Yes

4. Is the manuscript presented in an intelligible fashion and written in standard English?

Reviewer #1: Yes

Reviewer #2: Yes

5. Review Comments to the Author

Reviewer #1: In the present study, the authors demonstrate the critical role of the PHD2 pathway in the hypoxic signaling cascade. Hypoxic signaling activates the AMPK system by directly phosphorylating the PHD2 pathway, leading to a significant endogenous hypoxic protective effect. The study's objective and rationale are well-described, and the authors provide a clear explanation of the methods used. The results obtained are significant.

Here are a few minor suggestions for improvement:

The abbreviation "DMEM" is not explained on page 4, line 86.

The abbreviation "XF" is not explained on page 6, line 126.

In the References section, the word "References" should be written before listing the references.

In reference number 7, the journal name is written in all capital letters. It should be formatted similarly to the other references.

In Figure 4, the labels for "sh-PHD2" and "sh-PDH2+8μM Dorsomorphin" are not written properly in the histogram.

Reviewer #2: In current study, Xie et al. did a detail study of Silencing of topical proline hydroxylase domain 2 promotes the healing of rat diabetic wounds by phosphorylating AMPK. In my opinion, the study design is good and well-illustrated. However following points should be taken into consideration in the present form of this manuscript.

1. To achieve effective suppression of PHD2, authors should possess a comprehensive understanding of transcriptomics through thorough and high-throughput study.

2. The authors should emphasize the distinct pathways that undergo changes due to sh-PHD2. These pathways should be elaborated upon in greater detail within the discussion section.

3. The authors should verify that the figures in the panel maintain an optimal resolution for the imaging of Figure 3, Figure 5, and Figure 6.

4. Authors should ensure that the gene names should be written in italics.

6. PLOS authors have the option to publish the peer review history of their article (what does this mean?). If published, this will include your full peer review and any attached files.

Reviewer #1: **Yes: **Rajnesh Kumari Yadav

Reviewer #2: No

---

## [Author Response · Author response to Decision Letter 0]

4 Oct 2023

Dear Editor, Dear reviewers,

Thank you for your letter dated September 1. We were pleased to know that our work was rated as potentially acceptable for publication in Journal, subject to adequate revision. We thank the reviewers for the time and effort that they have put into reviewing the previous version of the manuscript. Their suggestions have enabled us to improve our work. Based on the instructions provided in your letter, we uploaded the file of the revised manuscript. Accordingly, we have uploaded a copy of the original manuscript with all the changes highlighted by using the track changes mode in Word. Appended to this letter is our point-by-point response to the comments raised by the reviewers. The comments are reproduced and our responses are given directly afterward in a different color (red). We would like also to thank you for allowing us to resubmit a revised copy of the manuscript.

We hope that the revised manuscript is accepted for publication in“PLOS ONE” . 

 Sincerely, Hong Yan

Responses to journals:

Responses:We have revised the manuscript according to your template standard when we started to submit the manuscript, if there is any deficiency, we hope that you can clearly point it out, and we will make changes to it as soon as possible.

2.To comply with PLOS ONE submissions requirements, in your Methods section, please provide additional information regarding the experiments involving animals and ensure you have included details on (1) methods of sacrifice, (2) efforts to alleviate suffering.

Responses:We have added methods of sacrifice and efforts to alleviate sufferingon page 4, line 85 of the manuscript. if there are any deficiencies, we hope that you will let us know and we will perfect them at the first opportunity.

"Tthe authors who received the award is Mr Yan . The full name of the funder is (Luzhou-Medical University) Cooperation Project,the grant number is No. 2019LZXNYDZ08.The URL of funder website is http://luzhou.tccxfw.com .The sponsor was involved in the design of this study."

Responses:For contributions made by funders, we have added information in Cover Latter.

Responses:We have uploaded the minimum dataset as support information.

Responses:We have uploaded all gel results as support information.

6.Please upload a copy of Supporting Information Figure/Table/etc. Fig.1-7 which you refer to in your text on page 22.

Responses:We have uploaded all the information corresponding to Fig.1-7 as supporting information.

7.Please review your reference list to ensure that it is complete and correct. If you have cited papers that have been retracted, please include the rationale for doing so in the manuscript text, or remove these references and replace them with relevant current references. Any changes to the reference list should be mentioned in the rebuttal letter that accompanies your revised manuscript. If you need to cite a retracted article, indicate the article’s retracted status in the References list and also include a citation and full reference for the retraction notice.

Responses:In the subsequent discussion, we have added further information on the impact of altered PHD2 on other relevant pathways, and therefore cited three additional papers, references 37, 38, and 39.All references are complete and correct, and all references are formatted and inserted in the format required by the journals.

Responses to reviewer 1:

1、The abbreviation "DMEM" is not explained on page 4, line 86.

Responses:We have added to the description of "DMEM" on page 4, line 86 of the manuscript, with the following explanation: Dulbecco's Modified Eagle Medium Thank you for your suggestion!

2、The abbreviation "XF" is not explained on page 6, line 126.

Responses:We have explained "XF" on page 6, line 126 of the manuscript, specifically: Extrancellular Flux .Thank you for your suggestion!

3、In the References section, the word "References" should be written before listing the references.

Responses:We've added this detail before the reference, thanks for the suggestion!

4、In reference number 7, the journal name is written in all capital letters. It should be formatted similarly to the other references.

Responses:We have re-corrected the formatting of all references as required by the journal, thank you for your suggestion!

5、In Figure 4, the labels for "sh-PHD2" and "sh-PDH2+8μM Dorsomorphin" are not written properly in the histogram.

Responses:As Figure 4 is not standardised enough, we have recreated Figure 4 and re-uploaded it, thank you for your suggestion.

Responses to reviewer 2:

1.To achieve effective suppression of PHD2, authors should possess a comprehensive understanding of transcriptomics through thorough and high-throughput study.

Responses:Due to the time problem, we can't improve the PHD2 related high throughput research for the time being, in the future our group will continue to improve the related research for PHD2, thank you for your suggestion!

2.The authors should emphasize the distinct pathways that undergo changes due to sh-PHD2. These pathways should be elaborated upon in greater detail within the discussion section.

Responses:On page 16, line 350 of the manuscript, we provide additional elaboration in response to the other pathways that are altered due to sh-PHD2, and thank you for the suggestion!

3.The authors should verify that the figures in the panel maintain an optimal resolution for the imaging of Figure 3, Figure 5, and Figure 6.

Responses:When uploading Figure 3, Figure 5, and Figure 6., we are sure that it is already the best resolution, we are not sure why the resolution has dropped in the finished manuscript, we will upload all the figures again, if the result is still not good, we hope that you can contact us, thank you for your suggestion!

4.Authors should ensure that the gene names should be written in italics.

Responses:We have gone through the manuscript again and changed all gene names to italics, specifically: manuscript page 8, line 174; manuscript page 17, line 360.

---

## [Decision Letter · Decision Letter 1]

5 Nov 2023

Silencing of topical proline hydroxylase domain 2 promotes the healing of rat diabetic wounds by phosphorylating AMPK

PONE-D-23-10650R1

Dear Dr. Yan,

We’re pleased to inform you that your manuscript has been judged scientifically suitable for publication and will be formally accepted for publication once it meets all outstanding technical requirements.

Kind regards,

Kanhaiya Singh, Ph.D

Academic Editor

PLOS ONE

Additional Editor Comments (optional):

Reviewers' comments:

Reviewer's Responses to Questions

**Comments to the Author**

1. If the authors have adequately addressed your comments raised in a previous round of review and you feel that this manuscript is now acceptable for publication, you may indicate that here to bypass the “Comments to the Author” section, enter your conflict of interest statement in the “Confidential to Editor” section, and submit your "Accept" recommendation.

Reviewer #1: All comments have been addressed

Reviewer #2: All comments have been addressed

2. Is the manuscript technically sound, and do the data support the conclusions?

Reviewer #1: Yes

Reviewer #2: Yes

3. Has the statistical analysis been performed appropriately and rigorously? 

Reviewer #1: Yes

Reviewer #2: Yes

4. Have the authors made all data underlying the findings in their manuscript fully available?

Reviewer #1: Yes

Reviewer #2: Yes

5. Is the manuscript presented in an intelligible fashion and written in standard English?

Reviewer #1: Yes

Reviewer #2: Yes

6. Review Comments to the Author

Reviewer #1: Author successfully addressed all the comments.

Author successfully addressed all the comments given by me as a reviewer.

Reviewer #2: (No Response)

7. PLOS authors have the option to publish the peer review history of their article (what does this mean?). If published, this will include your full peer review and any attached files.

Reviewer #1: **Yes: **Rajnesh Kumari Yadav

Reviewer #2: No

---

## [Editor Report · Acceptance letter]

16 Nov 2023

PONE-D-23-10650R1 

Silencing of topical proline hydroxylase domain 2 promotes the healing of rat diabetic wounds by phosphorylating AMPK 

Dear Dr. Yan:

I'm pleased to inform you that your manuscript has been deemed suitable for publication in PLOS ONE. Congratulations! Your manuscript is now with our production department. 

Kind regards, 

on behalf of

Dr. Kanhaiya Singh 

Academic Editor

PLOS ONE